# Small-sided games: An umbrella review of systematic reviews and meta-analyses

**Filipe Manuel Clemente**[1,2]*, **José Afonso**[3], **Hugo Sarmento**[4]

**1** Escola Superior Desporto e Lazer, Instituto Politécnico de Viana do Castelo, Rua Escola Industrial e Comercial de Nun'Álvares, Viana do Castelo, Portugal, **2** Instituto de Telecomunicações, Delegação da Covilhã, Lisboa, Portugal, **3** Centre for Research, Education, Innovation and Intervention in Sport, Faculty of Sport of the University of Porto, Porto, Portugal, **4** Research Unit for Sport and Physical Activity, Faculty of Sport Sciences and Physical Education, University of Coimbra, Coimbra, Portugal

* filipe.clemente5@gmail.com

## Abstract

### Objective

This umbrella review was conducted to summarize the evidence and qualify the methodological quality of SR and SRMA published on small-sided games in team ball sports.

### Methods

A systematic review of Web of Science, PubMed, Cochrane Library, Scopus, and SPORT-Discus databases was performed according to the Preferred Reporting Items for Systematic Reviews and Meta-Analyses (PRISMA) guidelines.

### Results

From the 176 studies initially identified, 12 (eight SR and four SRMA) were fully reviewed, and their outcome measures were extracted and analyzed. Methodological quality (with the use of AMSTAR-2) revealed that seven reviews had low quality and five had critically low quality. Two major types of effects of SSGs were observed: (i) short-term acute effects and (ii) long-term adaptations. Four broad dimensions of analysis were found: (i) physiological demands (internal load); (ii) physical demands (external load) or fitness status; (iii) technical actions; and (iv) tactical behavior and collective organization. The psychological domain was reduced to an analysis of enjoyment. The main findings from this umbrella review revealed that SSGs present positive effects in improving aerobic capacity and tactical/technical behaviors, while neuromuscular adaptations present more heterogeneous findings. Factors such as sex, age group, expertise, skill level, or fitness status are also determinants of some acute effects and adaptations.

### Conclusion

The current umbrella review allowed to identify that most of the systematic review and meta-analysis conducted in SSGs presents low methodological quality considering the standards. Most of the systematic reviews included in this umbrella revealed that task constraints significantly change the acute responses in exercise, while SSGs are effective in improving

**Data Availability Statement:** This is a systematic review without use of original data. However, all relevant information needed to replicate the findings of this study are included in the manuscript and supporting information files.

**Funding:** This work is funded by Fundação para a Ciência e Tecnologia/ Ministério da Ciência, Tecnologia e Ensino Superior through national funds and when applicable co-funded EU funds under the project UIDB/50008/2020." Hugo Sarmento gratefully acknowledge the support of a Spanish government subproject Integration ways between qualitative and quantitative data, multiple case development, and synthesis review as main axis for an innovative future in physical activity and sports research [PGC2018-098742-B-C31] (Ministerio de Economía y Competitividad, Programa Estatal de Generación de Conocimiento y Fortalecimiento Científico y Tecnológico del Sistema I+D+i), that is part of the coordinated project 'New approach of research in physical activity 28 and sport from mixed methods perspective (NARPAS_MM) [SPGC201800X098742CV0]'. No other specific sources of funding were used to assist in the preparation of this article.

**Competing interests:** The authors have declared that no competing interests exist.

aerobic capacity. Future original studies in this topic should improve the methodological quality and improve the experimental study designs for assessing changes in tactical/technical skills.

## 1. Introduction

Drill-based activities are part of the training resources used by coaches in team ball sports [1]. Among others, small-sided games (SSGs), also known as small-sided or conditioned games, are constrained game-based drills that change the structural dynamic of the formal match [2]. These games are very popular in team ball sports, namely because they are conceived by coaches to augment the perception of the players for specific tactical/technical issues while some of these challenges promote variations in the physiological and physical stimuli [3]. Besides the variations occurring during these games, SSGs elicits an intensification in physiological stimulus occurring during the exercise, thus being typically used as forms of high-intensity interval training [4, 5] with possible consequences for biological adaptations in the medium-to-long term [6]. The consistent use of SSGs possibly may also help in improving specific technical skills or tactical behaviors by considering the pedagogical principles worked out during these drills [7, 8].

The design of SSGs is closely related to the management of task constraints [2]. For a typical task, constraints include–but are not limited to–changes in format (number of players involved in the game and numerical relationships), pitch configuration (e.g., individual area per player, width-to-length ratio, the shape of the pitch), scoring method (e.g., using or not using goals or targets, having or not having goalkeepers, changing the objective for scoring), permitted actions (e.g., limitation in ball touches or movements), tactical and strategic missions (e.g., specific instructions), and/or training regimen (e.g., sets, repetitions, work-to-rest ratio) [9, 10]. The interaction between these task constraints promotes acute effects in biological responses to the exercise (i.e., internal load) and physical demands (i.e., external load), as well as variations in tactical behaviors or technical actions [11].

However, the use of SSGs as part of structured training programs may also offer possibilities for adaptations in fitness status or tactical/technical dimensions [7]. In fact, SSGs have been regularly compared with running-based drills, mainly considering the adaptations in fitness status [6]. Nevertheless, the magnitude of adaptations or acute effects can be affected by different factors, such as age group and expertise levels, sex of the players, baseline level in fitness status or technical/tactical dimensions, or even other mental and psychological aspects [12]. Furthermore, it is still not clear how the applications of SSGs should be framed into structured, longer-term periods or what their long-term adaptations may be [13].

The last two decades have shown a proliferation of publications of empirical studies, and in the last decade, systematic reviews (SRs) of small-sided games have emerged. The progressive number of SRs about the topic provides an opportunity to synthesize and summarize the main evidence from multiple research syntheses. In fact, despite using the same topic (SSGs) different methodological approaches conducts different findings in SRs. For that reason, it is important to assess the methodological quality of the current SRs about SSGs, as well as, provide an opportunity to synthesize the main evidence and the lack of evidence that must provide new lines for future research. An umbrella review may offer a rapid review of the evidence and provide an overall examination of the body of information that is available for the topic of SSGs [14]. In fact, an umbrella review will provide a clear opportunity to have a wide picture of the

consistency or not of evidence around SSGs [14]. Additionally, it will be possible to have a broader picture about different approaches conducted in SSGs.

Based on these reasons, the purposes of this article is to employ an umbrella review in SSGs that allows: (i) to systematically review available SR and systematic reviews with meta-analysis (SRMA) about SSGs in team ball sports; (ii) to qualify the methodological quality of these SR and SRMA, as well as to identify their strengths and limitations; and (iii) to summarize the main evidence presented in SR and SRMA, identify possible gaps in the literature, and provide recommendations for future research on SSGs. These objectives will help to provide a wide picture of the research conducted in SSGs, as well as, define new lines and opportunities for research. Will also help to provide relevant information about the consistency or not of specific evidence about SSGs. For that reason, it is expectable that this umbrella review reveals consistency or inconsistency of changing task conditions in specific internal and external load demands, as well as, in technical/tactical skills by summarizing the SRs. Additionally, it is expectable to provide information about consistency or not of SSGs for developing specific physical qualities or technical/tactical skills, by summarizing the evidence of meta-analysis about the topic.

## 2. Methods

The present umbrella review of SR and SRMA was conducted in accordance with the Preferred Reporting Items for Systematic Reviews and Meta-analyses (PRISMA) guidelines [15]. The protocol was registered with the International Platform of Registered Systematic Review and Meta-Analysis Protocols with the number 202080068 and the DOI number 10.27766/inplasy2020.8.0068.

### 2.1. Information sources

A comprehensive computerized search of the following electronic databases was performed: (i) Web of Science (all databases); (ii) Scopus; (iii) SPORTDiscus; (iv) PubMed; and (v) Cochrane library (Cochrane Database of Systematic Reviews). The search process for relevant publications had no restriction regarding year of publication and included SR and SRMA articles retrieved until 15th of August of 2020. The following search strings were employed: ("small-sided games" OR "sided-games" OR "drill-based games" OR "SSG" OR "conditioned games" OR "small-sided and conditioned games" OR "reduced games" OR "play formats") AND ("team sport" OR football OR soccer OR futsal OR handball OR volleyball OR basketball OR hockey OR rugby OR cricket OR "water polo" OR lacrosse OR softball OR korfball) AND ("systematic review" OR "meta-analysis"). Additionally, the reference lists of the studies retrieved were manually searched to identify potentially eligible studies not captured by the electronic searches. Finally, an external expert (Ph.D., assistant professor and ten years of publications in SSGs, with international publications in the topic) has been contacted in order to verify the final list of references included in this umbrella review in order to understand if there was any study that was not detected through our research. The inclusion and exclusion criteria were also provided to the external expert (as suggested by Cochrane's guidelines) [16], however, no information about which databases to consult or search strategies to use were provided aiming tot create a bias in the expert. This process (double-check of an external expert) is recommended by PRISMA guidelines and also is part of the assessment items in AMSTAR-2 [17]. The external expert did not add new articles for inclusion, thus confirming the accuracy of the initial search.

## 2.2. Eligibility criteria

The inclusion criteria for this umbrella review were as follows: (i) only SR or SRMA in SSGs (not limited to the type of study designs included in the SR or SRMA) in team ball sports; (ii) any SR or SRMA in SSGs that included outcomes related to an internal and external load (acute responses to SSGs), fitness variables (adaptations after a certain period of SSGs-based intervention) technical/tactical measures (acute responses or adaptations), psychological or pedagogical dimensions (acute responses or adaptations); and (iii) peer-reviewed SR and SRMA written in English that provided full-text. Studies were excluded on the basis that they: (i) were not SR or SRMA (e.g., narrative reviews, brief reviews, scoping reviews, empirical articles, methodological proposals); (ii) did not include relevant data in SSGs and/or team ball sports; (iii) were not fully written in English; and (iv) consisted of abstracts only, without accompanying full-texts.

The screening of the title, abstract and reference list of each study to locate potentially relevant studies was independently performed by two of the authors (FMC and HS). Additionally, they reviewed the full version of the included papers in detail to identify articles that met the selection criteria. A third author (JA) participated to resolve discrepancies regarding the selection process.

## 2.3. Data extraction

A data extraction was prepared in Microsoft Excel sheet (Microsoft Corporation, Readmon, WA, USA) in accordance with the Cochrane Consumers and Communication Review Group's data extraction template [18]. The Excel sheet was used to assess inclusion requirements and subsequently tested for all selected studies. The process was independently conducted by two of the authors (FMC and HS). Any disagreement regarding study eligibility was resolved by a third author (JA). Full text articles excluded, with reasons, were recorded. All the records were stored in the sheet.

## 2.4. Data items

The following information was extracted from the included SR and SRMA: (i) number of original articles included (n), age-group (youth, adults or both), sex (men, women or both), competitive level (if available), design (SR or SRMA) and type of original articles included (experimental, observational analytic or both); (ii) identification of the effects (acute or adaptations), dimension of analysis (internal load, external load, technical, tactical, recovery/fatigue/readiness, psychological), outcomes explored, and main findings.

## 2.5. Assessment of methodological quality

The Assessing the Methodological Quality of Systematic Reviews (AMSTAR-2) tool [17] was used to assess the methodological quality of the SR and SRMA included in this umbrella-review. The AMSTAR-2 it is a rating system that classify all reviews' quality level into critically low, low, moderate and high. The system classifies 16 items, namely [17]: (i) information about the use of PICO; (ii) statement about the methods made before conducting research; (iii) explanation for inclusion of study designs; (iv) use of comprehensive literature search strategy; (v) study selection made in duplicate; (vi) data extraction in duplicate; (vii) list of excluded studies and reasons; (viii) describe included studies in detail; (ix) assessing the risk of bias; (x) report sources of funding for the included studies; (xi) appropriate statistical methods used in the meta-analysis; (xii) assess the potential impact of risk of bias on the results; (xiii) consider the risk of bias in primary outcomes when interpreting/discussing the results; (xiv)

appropriate explanation about heterogeneity observed in the results; (xv) conduct an adequate investigation of publication bias and discuss its likely impact on the results, and (xvi) report potential sources of conflict of interest. The quality of each eligible SR and SRMA was analysed by two researchers (FMC and HS) independently. If the assessment was unclear, consensus was achieved with the help of a third author (JA).

The overall confidence in the results of the systematic reviews proposed by the AMSTAR-2 tool was defined as: (1) high—No, or one non-critical weakness: the systematic review provides an accurate and comprehensive summary of the results; (2) moderate—more than one non-critical weakness but no critical flaws: the systematic review provides an accurate summary of the results; (3) low—one critical flaw, with or without non-critical weaknesses: the systematic review may not provide an accurate and comprehensive summary of the results; (4) critically low—more than one critical flaw, with or with-out non-critical weaknesses: the review should not be relied on to provide an accurate and comprehensive summary of the results [17].

Data were retrieved by two authors (FMC and HS) and checked by a third author (JA). A specifically designed template for data extraction was developed. For each included SR or SRMA, the following items were extracted: study citation details, purpose of the study and context of analysis, type of analysis, match context, individual and environmental constraints and, main outcomes. Since the data were descriptively reported in this umbrella review, there was no statistical analysis.

## 3. Results

### 3.1. Study identification and selection

The searching of databases identified a total of 176 titles. These studies were then exported to reference manager software (EndNote™ X9, Clarivate Analytics, Philadelphia, PA, USA). Duplicates (88 references) were subsequently removed either automatically or manually. The remaining 88 articles were screened for their relevance based on titles and abstracts, resulting in the removal of a further 65 studies. Following the screening procedure, 23 articles were selected for in depth reading and analysis. After reading full texts, a further 11 studies were excluded due to not being SR or SRMA (n = 8) and not being written in English (n = 3) (Fig 1). From the 12 papers included in this umbrella review, 8 were SR and 4 were SRMA. The chronological analysis of the articles considered in this umbrella review revealed the recent developments in this area of research, highlighting that 8 of 12 articles (67%) were published between 2019 and 2020 (last two years) and the oldest SR were published in 2011 (9 years ago).

### 3.2. Study characteristics and qualitative synthesis

The characteristics of the 12 SR and SRMA included in the umbrella review can be found in Table 1, while the summary of the main evidence found in each review can be found in Table 2. Notably, most studies included in the reviews lasted between six and eight weeks of training, sometimes up to 10 or 12 weeks, occasionally under six weeks, and one odd study analyzed by [19] lasted 18 weeks. Most studies used single groups or parallel, non-randomized groups. In the reviews conducted on soccer [9, 19], only two research articles in each included randomization. In the review by [20], only one article provided randomization. Seven reviews did not report on randomization of the included studies or explicitly used observational studies (i.e., single-group) only [21]. One review even explicitly acknowledged the lack of randomization of original research papers as a major shortcoming [22]. The exception was the review on team sports by [23], in which 14 of the 16 studies had followed randomized interventions.

Of the included articles, eight focused on soccer [6, 21, 22, 24–28], one focused on basketball [29], and three examined team ball sports [20, 23, 30]. This reveals that most of the existing

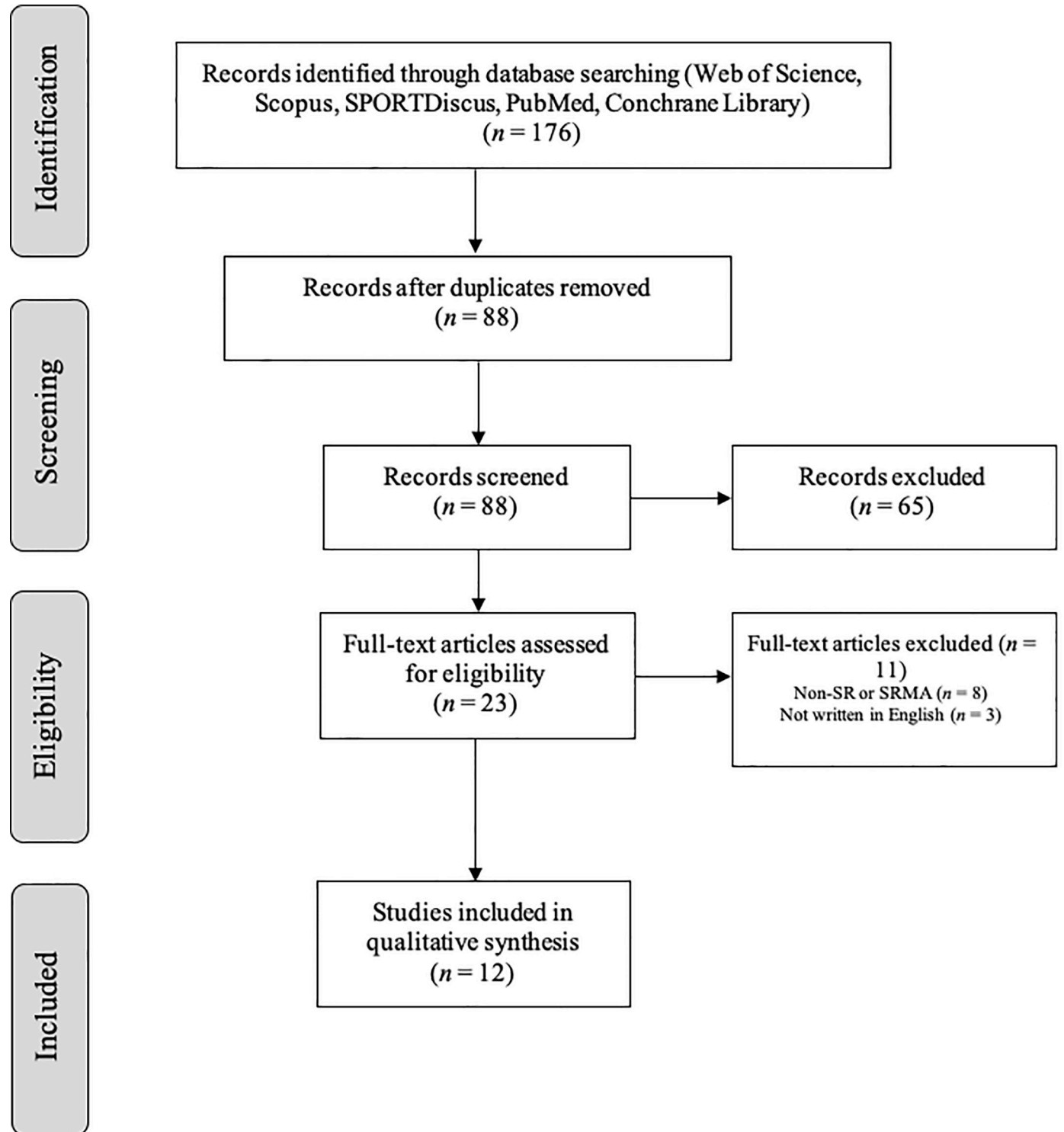

**Fig 1. PRISMA flow diagram highlighting the selection process for the studies included in the umbrella-review.**

**Table 1. Summary of SR and SRMA characteristics.**

| Reference | Team sport | Type of review (SR, SRMA) | Original studies included (N) | Type of included studies (E, OA) | Age group (Y, Ad) and sex (M, W) | Analysis of acute effects | | | | | Analysis of adaptations | |
|---|---|---|---|---|---|---|---|---|---|---|---|---|
| | | | | | | Analysis of internal load | Analysis of external load | Analysis of tactical dimension | Analysis of technical dimension | Other analysis (recovery, fatigue, psychological) | Fitness status | Technical & tactical |
| Bujalance-Moreno et al. [19] | Soccer | SR | 53 | E and OA | Y and Ad | Yes | Yes | Yes | Yes | Yes | Yes | No |
| | | | | | M | | | | | | | |
| Clemente et al. [22] | Soccer | SR | 34 | OA | Y and Ad | No | No | Yes | No | No | No | No |
| | | | | | ND | | | | | | | |
| Clemente & Sarmento [36] | Soccer | SR | 37 | E and OA | Y and Ad | No | No | No | Yes | No | No | No |
| | | | | | ND | | | | | | | |
| Fernández-Espínola et al. [30] | Team sports | SR | 47 | OA | Y | No | No | Yes | Yes | No | No | No |
| | | | | | M and W | | | | | | | |
| Hammami et al. [7] | Team sports | SRMA | 16 | E and OA | Y and Ad | No | No | No | No | No | Yes | Yes |
| | | | | | M and W | | | | | | | |
| Hill-Haas et al. [26] | Soccer | SR | ND | E and OA | Y and Ad | Yes | No | No | No | No | Yes | No |
| | | | | | M and W | | | | | | | |
| Kunz et al. [27] | Soccer | SRMA | 9 | E | Y | No | No | No | No | No | Yes | No |
| | | | | | M | | | | | | | |
| Moran et al. [6] | Soccer | SRMA | 8 | E | Y | No | No | No | No | No | Yes | No |
| | | | | | M | | | | | | | |
| Nygaard Falch et al. [20] | Team sports | SRMA | 74 | E | Y and Ad | No | No | No | No | No | Yes | No |
| | | | | | M and W | | | | | | | |
| O'Grady et al. [29] | Basketball | SR | 17 | OA | Y and Ad | Yes | Yes | No | No | No | No | No |
| | | | | | M and W | | | | | | | |
| Ometto et al. [21] | Soccer | SR | 24 | OA | Y and Ad | No | No | Yes | Yes | No | No | No |
| | | | | | ND | | | | | | | |
| Sarmento et al. [28] | Soccer | SR | 77 | OA | Y and Ad | Yes | Yes | Yes | Yes | Yes | No | No |
| | | | | | M and W | | | | | | | |

SR: systematic reviews; SRMA: systematic reviews and meta-analysis; E: experimental; OA: observational analytic; Y: youth; Ad: adults; M: men; W: women; ND: not defined.

systematized knowledge is derived from applications in soccer. Additionally, three of the reviews analyzed only men [6, 19, 27], six analyzed both sexes [9, 20, 23, 28–30], and the remaining three did not report participants' sex [21, 22, 25]. Regarding age groups, three reviews included only youth players [6, 27, 30], and the remaining nine included both youth and adults. Internal load monitoring was reported in only four reviews [9, 19, 28, 29]; external load monitoring was also reported in three of those four reviews [19, 28, 29] and none of the others, meaning that the load produced by SSGs is not well-known or, at least, not well-systematized in reviews.

In terms of dimensions of analysis, only five of the 12 reviews analyzed the tactical dimension [19, 21, 22, 28, 30], which was unexpected given that one of the main goals of SSGs is to promote the learning of game principles. Also, only five reviews analyzed the technical dimension [19, 21, 25, 28, 30], meaning that fewer than 50% of the studies have verified the impact of

**Table 2. Summary of main findings presented in the SR and SRMA.**

| Reference | Acute effects: structural design | Acute effects: training regimen | Acute effects: contextual factors | Adaptations: fitness status and tactical/ technical |
|---|---|---|---|---|
| Bujalance-Moreno et al. [19] | *Format*: lower number of players elicits greater %HR, RPE, and BLa. Moreover, changes in format also promote alterations in DC at different intensities. Smaller formats increase individual technical actions made by players. *Pitch configuration*: larger pitch sizes elicit greater physiological intensity. Larger pitches elicit increases in DC. *Marking*: type of marking influences physiological demands. *Scoring method*: SSGs with possession of the ball and without goalkeepers or small goals increase the intensity. Additionally, games with goals and goalkeepers reduce external load demands. Considering technical actions, extreme limitations (such as one ball touch) reduce the success in passes and duels. *Action restriction*: ball touch limitations elicit increases in BLa and RPE. It also increases DC at high intensities. | *Continuous vs. intermittent*: equivocal findings in RPE and % HRmax. Greater DC at high intensity in intermittent regimens. *Short vs. long bout duration*: shorter duration elicits lower % HRmax. However, shorter durations elicited greater DC at moderate intensities. *Active rest vs. passive*: active rest seems to contribute to greater physical demands. | *Competitive level*: amateur players perform less DC at high intensities than professionals. Succeeded passes and ball possession are also higher in professionals. | *Aerobic performance*: SSG-based program elicits significant improvements in different aerobic performance measures. *Neuromuscular performance*: limited effects of SSGs were found on sprinting. Drills with ball promoted better improvements in sprinting than SSGs. Multidirectional running had better effects on COD and linear sprinting than SSGs. Baseline levels (e.g., RSA) meaningfully influenced the improvements (or not) in some studies. |
| Clemente et al. [22] | *Format*: larger formats increased distances from teammates to the team's centroid, and movements become more regular. Additional players (floaters) contributed to increasing the playing space. Smaller formats promoted more individual tactical actions (e.g., penetration, delays), while larger formats promoted more collective behaviors (e.g., unity). Numerical imbalances make defensive teams more regular and organized. *Pitch configuration*: larger pitch sizes increased variability of actions and exploration. Additionally, using a small length-to-width ratio increased the exploration of the wings of the pitch, while restricted pitches increased the variability of movements. The use of pitch restrictions promoted regularities in the movements. *Scoring method*: the use of GK increases the length and width while attacking and makes closer players while defending. Games without goals (ball possession games) increase passing sequences. *Tactical/strategic mission*: defensive instructions contributed to increases in defensive actions and made the teammates closer, while offensive instructions increased the spread of the players. | - | *Age*: older players kept greater distances between the teams' centroids and explored the width more, while younger players explored more the longitudinal space. *Competitive level*: national level players (compared to regional ones) present greater dispersion and unpredictability in the moments while playing. *Fitness*: no meaningful influence of fitness status on behavior. *Psychological*: mental fatigue conducted to decreases in dyadic synchronization. However, players became more synchronized. | - |

(*Continued*)

**Table 2.** (Continued)

| Reference | Acute effects: structural design | Acute effects: training regimen | Acute effects: contextual factors | Adaptations: fitness status and tactical/technical |
|---|---|---|---|---|
| Clemente & Sarmento [36] | *Format*: smaller formats increase the number of technical actions. Unbalanced numerical relationships (e.g., using floaters) increased the success in technical actions, despite reducing the frequency of dribbles or duels. *Pitch configuration*: smaller relative pitch area increased frequencies of technical actions, while larger pitches contributed to longer ball possessions. Short pitches increased attacking finalization, and long pitches increased passes and ball possession. *Scoring method*: inclusion of goals and goalkeepers reduces success in passes and ball possession duration, while small goals or no goals increase the success rate of these actions. The use of goals on sides will increase the frequency of sideways passes and turns. *Action restriction*: free play increased duels. However, limited ball touches increase technical actions. Extreme limitations (such as 1-ball touch) decrease the success of passes. *Tactical/strategic mission*: the use of defensive instruction increased the frequencies of balls recovered, and offensive instructions increased the number of passes. | *Continuous vs. intermittent*: fewer goals occurred in a continuous regimen. *Short vs. long bout duration*: longer periods of recovery increased the total passes and succeeded passes. | *Age*: Older players perform more individual actions than younger ones. *Expertise*: experienced players perform longer offensive sequences while using passing and more touches. Non-experienced players opt more often for individual actions to solve attacks. *Fitness*: no meaningful influence of fitness status on technical actions. *Psychological*: Mental fatigue contributed to decreases in technical actions' success. | *Tactical/technical*: SSG-based programs contributed to improvements in technical skills. Specific constraints used in SSG-based programs produced different improvements in young players. |
| Fernández-Espinola et al. [30] | *Format*: smaller formats increase the number of technical actions. *Pitch configuration*: generally, high frequencies of technical actions in smaller pitches. Team width and length of both teams were greater in larger pitches. *Scoring method*: ball possession games led to more positional attacks and contacts with the ball. Standard games led to faster attacks, more individual actions, and fewer ball touches. Goal scoring games increased the success of decision making compared to ball possession games. | *Continuous vs. intermittent*: continuous regimen conducted to greater frequencies of dribbles. *Short vs. long bout duration*: no significant effects on technical actions. | *Age*: older participants showed higher levels of collective tactical behavior. Younger participants also presented shorter offensive sequences and made more individual actions. *Maturation*: no significant correlations with technical/tactical actions. *Psychological*: the presence of coaches conducted to decreases in the success of technical actions, such as passing. | - |
| Hammami et al. [7] | - | - | - | *Aerobic performance*: large beneficial effects (ES = 1.94) of SSGs were found in VO₂max. *Neuromuscular performance*: large beneficial effects of SSGs were found in agility and RSA (ES = 1.19). Moderate beneficial effects were found in the linear sprint (ES = -0.89), vertical jump ((ES = 0.68), and intermittent endurance (ES = 0.61). *Tactical/technical*: large positive effects of SSGs on specific sports tasks. *Psychological*: positive effects on enjoyment. *Contextual factors*: improvements in fitness were found independently of competitive level (amateur, elite, and professional). |

(*Continued*)

**Table 2.** (Continued)

| Reference | Acute effects: structural design | Acute effects: training regimen | Acute effects: contextual factors | Adaptations: fitness status and tactical/technical |
|---|---|---|---|---|
| Hill-Haas et al. [26] | *Format:* having fewer players elicits increases in HR, BLa, and RPE. No significant differences in unbalanced formats. Floater performed greater DC in smaller formats. *Pitch configuration:* increases in HR, RPE, and BLa on larger pitches. Concurrent increases in format and relative pitch area elicit lower physiological intensity. *Scoring method:* equivocal evidence about the effects of using or not using GK in internal load measures. *Action restriction:* a combination of SSGs and extra sprint effort led to greater external load but not to significant internal load changes. *Tactical/strategic mission:* player-to-player marking increases BLa and RPE. Direct supervising using encouragement elicits increases in RPE and HR. | *Continuous vs. intermittent:* intermittent regimens led to greater DC at different speed thresholds, while internal load was higher in continuous. *Short vs. long bout duration:* decreases in HR from medium to long bout duration were found. | - | *Aerobic performance:* SSGs and running-based HIIT were effective in improving aerobic performance with no changes between them. *Neuromuscular performance:* the jumping performance, COD, and multi-stage performance were improved by both SSGs and running-based HIIT, with no significant differences between them. |
| Kunz et al. [27] | - | - | - | *Aerobic performance:* comparison between running-based HIIT and SSGs were favorable (small magnitude) to running-based HIIT (ES = 0.15). However, trivial differences were found for the lactate threshold (ES = -0.01) and running economy (ES = 0.05). *Neuromuscular performance:* running-based HIIT promoted positive effects (ES = 0.15) in sprinting compared to SSGs. However, trivial differences were found in jumping performance (ES = 0.08). The RSA benefited mainly by running-based HIIT comparing to SSG (ES = 0.59). *Tactical/technical:* small negative effects (ES = -0.21) of HIIT in technical skills were found when compared to SSGs. |
| Moran et al. [6] | - | - | - | *Aerobic performance:* trivial differences (ES = 0.04) were found between SSGs and conventional endurance training in aerobic performance. Additionally, SSGs revealed a moderate positive effect (ES = 0.82) on aerobic performance. *Contextual factors:* programs longer than eight weeks and four sets per session were favorable to SSGs. |
| Nygaard Falch et al. [20] | - | - | - | *Neuromuscular performance:* The use of SSGs as specific velocity-oriented COD training intervention led to moderate to moderate improvements in COD test performance. |

*(Continued)*

**Table 2.** (Continued)

| Reference | Acute effects: structural design | Acute effects: training regimen | Acute effects: contextual factors | Adaptations: fitness status and tactical/technical |
|---|---|---|---|---|
| O'Grady et al. [29] | *Format:* fewer players increased the accumulated external load and technical actions compared to larger formats of play. Additionally, smaller formats meaningfully increased the internal load (RPE). *Pitch configuration:* larger pitch meaningfully increased the external load, while in smaller ones, there are decreases in DC and transitions, spending more time stationary or walking. Larger pitches also contributed to meaningful increases in RPE and HR during the games. *Action restriction:* no-stop games (no game clock stoppages) conducted to increases in external load demands, RPE, and %HRmax. Exclusion of dribbling possibly increases the intensity of cutting and running movements, as well as technical actions, such as the pass. *Tactical/strategic mission:* man-to-man marking may explain decreases in external load demands. | *Continuous vs. intermittent:* non-consistent findings suggest that internal load (RPE and %HRmax) was higher in continuous regimens. *Short vs. long bout duration:* non-consistent findings suggest that short-intermittent games elicited higher external load. | - | - |
| Ometto et al. [21] | *Format:* numerical superiority will increase tactical space behaviors, defensive coverage, concentration, and defensive unity while increasing the distance between teammates and creating more time to create technical actions. Numerical inferiority may conduct to more penetrations, while increasing time in lateral corridors and dribbling actions. *Pitch configuration:* larger pitches will increase tactical balance behaviors while increasing distance between teammates and the area occupied by the team. However, smaller pitches will increase the frequency of technical actions and tactical behaviors as concentration or defensive unit. *Scoring method:* having a greater number of targets will also increase the distance between teammates and the time spent in lateral corridors. More targets will increase maintenance of ball possession and proximity between players, while smaller targets will increase penetration, space, offensive unit, or concentration tactical principles. | - | - | - |

*(Continued)*

**Table 2.** (Continued)

| Reference | Acute effects: structural design | Acute effects: training regimen | Acute effects: contextual factors | Adaptations: fitness status and tactical/technical |
|---|---|---|---|---|
| Sarmento et al. [28] | *Format:* smaller formats increase physiological intensity and frequency of technical actions. Equivocal findings regarding formats and physical demands. Larger formats will increase tactical behaviors and their range. Unbalanced teams promoted smaller distances between teammates. *Pitch configuration:* larger pitches (and individual playing areas) increase physiological intensity. However, smaller pitches led to an increase in the frequency of technical actions and to a smaller distance between players and between teams. *Scoring method:* small goals or ball possession games elicit increases in physiological responses compared to games with goalkeepers. *Action restriction:* limiting ball touches increases the physiological responses during games compared to free play. However, equivocal findings were found considering the effects on external load. Free play elicits a higher frequency of passing success or number of duels. *Tactical/strategic mission:* man-to-man marking increased physiological response to the exercise. Verbal encouragement provided by coaches increased the HR, BLa, and RPE during games. | *Continuous vs. intermittent:* continuous games were not different from intermittent in HR. However, the increased the physical demand. *Short vs. long bout duration:* decreases in HR were found in long bouts. | *Age:* relative age effect played an important role in tactical behaviors played in SSGs. Better defensive behaviors were performed by players who were born at the beginning of the year. *Expertise:* talented players were more successful in SSGs and covered greater distances at high intensity. | - |

HR: heart rate; HRmax: maximal heart rate; BLa: blood lactate; DC: distance covered; RSA: repeated sprint ability; GK: goalkeeper; VO$_2$max: maximal oxygen uptake; HIIT: high-intensity interval training; COD: change of direction; ES: effect size.

**Table 3. Short-term, acute effects and adaptations of SSGs: Summary table.**

| Constraints | Internal load | External load | Tactical behavior | Technical actions |
|---|---|---|---|---|
| Format | Consistent findings revealing increases in internal load measures during smaller formats. | Inconsistent findings revealing increases of external load in smaller formats. | Inconsistent findings revealing increases in range of tactical behaviors during larger formats. Consistent findings revealing compactness of players while defending in unbalanced games. | Consistent findings revealing increases in the frequencies of technical actions in smaller formats. |
| Pitch configuration | Consistent findings revealing increases in internal load measures during larger pitches. | Consistent findings revealing increases in external load measures during larger pitches. | Consistent findings revealing increases in space between teammates and spatial exploration in larger pitches. | Consistent findings revealing increases in the frequency of technical actions in smaller pitches. |
| Scoring method | Consistent findings supporting the idea that using GK reduces the internal load. | Consistent findings supporting the idea that using GK reduces the external load. | Inconsistent findings revealing that greater number of targets will increase distance between teammates and the time spent in lateral corridors. | Consistent findings revealing increases in passing during ball possession games. |
| Action restriction | Consistent findings revealing increases on internal load promoted by limitations in ball touches. | Inconsistent findings revealing effects of ball touches limitations in external load. | No evidence found. | Consistent findings revealing increases on success of technical actions during free play. |
| Tactical/ strategic mission | Consistent findings suggesting the intensification of internal load in man-to-man defensive marking. Consistent findings revealing that encouragement increases the internal load during games. | Inconsistent findings about the effects of type of defensive marking in external load. | Consistent findings revealing the effects of specific verbal instruction (e.g., defensive or offensive) on compactness and spread of the players. | Consistent findings revealing the effects of specific verbal instruction (e.g., defensive or offensive) on technical actions performed. |
| Training regimen | Inconsistent findings about the effects of continuous and intermittent regimens on internal load. | Consistent findings suggesting the intensification of external load in intermittent regimens. Consistent findings suggesting the intensification of external load in short bouts comparing to long bout. | No evidence found. | Inconsistent findings about the effects of bout duration in technical actions. |
| Contextual factors | No evidence found. | Consistent findings revealing that better competitive level increase the external load during the games. | Consistent findings supporting the idea that older and more expert players perform better tactical behaviors in games and explore more the pitch. | Consistent findings revealing that mental fatigue conduct to decreases in technical actions success. |

Consistent: consensus of the reviews into present a given tendency; Inconsistent: non-consensus of the reviews or equivocal findings reported by the reviews regarding the tendency of the evidence; GK: goalkeeper.

SSGs on skills. Among the SR and SRMA analyzed, six exclusively focused on reporting acute effects [21, 22, 25, 28–30], four of them only on longer-term adaptations [6, 20, 23, 27], while the remaining summarized both acute effects and adaptations [9, 19]. From those exclusively focused on acute effects, one exclusively focused on the tactical dimension [22], one focused only on the technical dimension [25], and the remaining works combined more than one of

**Table 4. Longer-term adaptations to SSG-based programs: Summary table.**

| | Aerobic performance | Neuromuscular performance | Tactical/technical skills | Psychological dimension |
|---|---|---|---|---|
| Adaptations to SSG-based programs | Consistent evidence for improving VO₂max or aerobic performance in field-bases tests. | Inconsistent evidence for improving jumping performance, linear sprinting, RSA and COD ability. | Consistent evidence for improving specific tactical/ technical skills. | Consistent evidence for improving the enjoyment. |

Consistent: consensus of the reviews into present a given tendency; Inconsistent: non-consensus of the reviews or equivocal findings reported by the reviews regarding the tendency of the evidence; VO₂max: maximal oxygen uptake; RSA: repeated sprint ability; COD: change of direction.

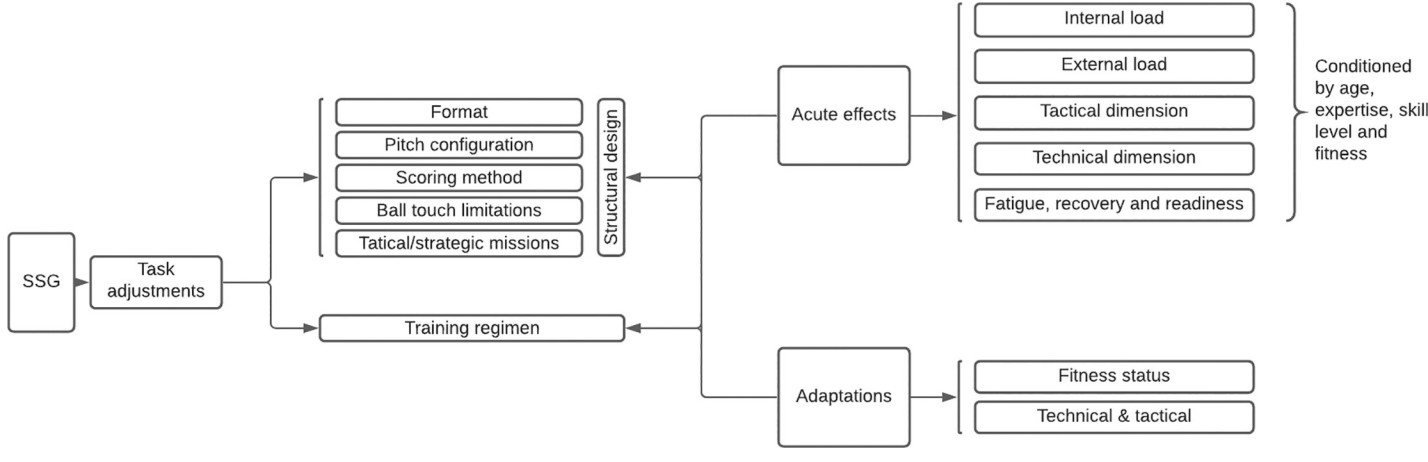

**Fig 2. Main small-sided games topics.**

these dimensions. The psychological adaptations were limited to acute enjoyment of engaging with SSGs.

All of the reviews exclusively focusing on longer-term adaptations have only reported variations in fitness status [6, 7, 20, 27], and therefore, the long-term tactical, technical, and psychological adaptations are not known. Combined with a lack of longitudinal randomized trials, the actual long-term adaptations and learning promoted by SSGs are currently unknown. Table 3 presents a synthesis of the short-term, acute effects and adaptations of SSGs, while Table 4 synthesizes the long-term adaptations.

A conceptual overview elaborated by the authors of this umbrella review can be seen in Fig 2. Hopefully, this overview helps in systematizing the complexity of the field and presenting it in an intelligible manner.

### 3.3. Methodological quality

The overall methodological quality of the 24 included papers is summarized in Table 5. Based on the overall confidence rating obtained through AMSTAR-2 [17], we concluded that the

**Table 5. AMSTAR 2 assessment of each included SR or SRMA.**

| Study | AMSTAR 2 –ITENS | | | | | | | | | | | | | | | | Overall Items |
|---|---|---|---|---|---|---|---|---|---|---|---|---|---|---|---|---|---|
| | 1 | 2 | 3 | 4 | 5 | 6 | 7 | 8 | 9 | 10 | 11 | 12 | 13 | 14 | 15 | 16 | |
| Bujalance-Moreno et al. [19] | Yes | No | Yes | Yes | Yes | No | No | PY | PY | No | No MA | No MA | Yes | No | No MA | Yes | Critically low |
| Clemente et al. [22] | Yes | No | No | Yes | Yes | Yes | Yes | PY | PY | No | No MA | No MA | Yes | No | No MA | Yes | Low |
| Clemente & Sarmento [36] | Yes | No | No | Yes | Yes | Yes | Yes | PY | PY | No | No MA | No MA | Yes | No | No MA | Yes | Low |
| Fernández-Espínola et al. [30] | Yes | No | No | Yes | No | No | Yes | Yes | Yes | No | No MA | No MA | Yes | No | No MA | Yes | Low |
| Hammami et al. [7] | Yes | No | No | Yes | No | No | No | Yes | PY | No | Yes | Yes | Yes | Yes | Yes | Yes | Critically low |
| Hill-Haas et al. [26] | Yes | No | Yes | Yes | No | No | No | Yes | No | No | No MA | No MA | No | No | No MA | Yes | Critically low |
| Kunz et al. [27] | Yes | No | Yes | Yes | No | Yes | Yes | Yes | Yes | No | Yes | Yes | Yes | Yes | Yes | Yes | Low |
| Moran et al. [6] | Yes | No | Yes | Yes | No | Yes | Yes | Yes | Yes | No | Yes | Yes | Yes | Yes | Yes | Yes | Low |
| Nygaard Falch et al. [20] | Yes | No | Yes | Yes | No | No | Yes | Yes | No | No | No | No | No | No | No | Yes | Critically low |
| O'Grady et al. [29] | Yes | No | Yes | Yes | Yes | Yes | Yes | Yes | Yes | No | No MA | No MA | Yes | No | No MA | Yes | Low |
| Ometto et al. [21] | Yes | No | No | Yes | Yes | No | Yes | PY | No | No | No MA | No MA | No | No | No MA | Yes | Critically low |
| Sarmento et al. [28] | Yes | No | No | Yes | Yes | Yes | Yes | PY | PY | No | No MA | No MA | Yes | Yes | No MA | Yes | Low |

overall confidence in the results of five reviews was rated as "critically low" and seven reviews were rated as "low." Considering the recent development of research on small-sided games in team sports, we decided not to exclude any of the studies based on their quality assessment because we believe that the proposal for methodological improvements in conducting systematic reviews in this area of research is of relevant importance at this stage. In addition, we consider that the aspects underlying obtaining the scores mentioned above do not significantly question the results presented in the studies included in this review. Lastly, it has been found that similar results were obtained for other areas of research when applying the AMSTAR-2 tool [31].

One factor that contributed to this low level of confidence was related to the lack of protocol registration in all of the analyzed systematic reviews and meta-analyses. In this sense, we recommend the registration of protocols before starting the reviews in one of the available platforms (e.g., Campbell, Cochrane, Inplasy, Open Science Framework, PROSPERO, among others). Importantly, all of the included studies were performed after the publication of the PRISMA statement in 2009 [32]. Additionally, in 50% of the studies, the authors did not explain their criteria for selection of the study designs and did not perform the study selection and extraction data in duplicates, which warrants concerns related to the rigor of the reviews due to a lack of error-correction methods.

Moreover, it should be noted that around 25% of the included papers did not apply a satisfactory technique for assessing the risk of bias (RoB) of the individual studies that were included in the review. However, the RoB of individual studies plays a major role when interpreting and discussing their results and may, therefore, profoundly impact the conclusions derived from the reviews, both qualitatively and quantitatively. Additionally, when heterogeneity is observed in the results, authors should provide a satisfactory explanation and critical discussion for it. Our umbrella review highlights the need to improve the quality of systematic reviews in this field of research. Otherwise, consensus and guidelines may be established based on unsubstantiated or fragile conclusions.

## 4. Discussion

The present umbrella review intended to summarize evidence related to (and to characterize the current status of) the SR and SRMA of SSGs. It was apparent there was a clear tendency for research in soccer (eight SR and/or SRMA) compared to basketball (one study) and team ball sports (three studies). However, despite the tendency of researchers to systematically review research in a single team sport, two broad dimensions of analysis in SSG studies were found in this umbrella review: (i) acute effects during games and (ii) longer-term adaptations promoted by SSG-based programs, although these were mostly non-randomized protocols.

Considering these dimensions, the present discussion is structured around the acute effects at the physiological/physical levels (internal and external loads) and the tactical/technical and psychological levels. The discussion also considers long-term adaptations at the fitness, tactical/technical, and psychological levels. For each main dimension, a discussion about the effects of constraints and contextual factors will be performed. After that, a summary is provided that encompasses the main findings and practical implications. A sub-section of the current research's limitations and future research directions for SSGs is also given.

### 4.1. Acute effects of SSGs: Internal and external loads

This current umbrella review showed that among the 12 articles summarizing acute effects, one exclusively centered on internal load (i.e., biological responses to the demands imposed by the SSG) [26], and three included internal and external loads [10, 19, 29]. Within internal load,

the most common measures analyzed were the heart rate (HR) during exercise, rate of perceived exertion (RPE) after exercise, and blood lactate concentration (BLa). For external load measures, the most recurrent measures were the distances covered at different speed thresholds.

Considering the acute effects of SSGs on internal load, it was found that different task constraints seem to work independently and/or concurrently to promote significant variations in the biological response [10, 19, 26, 29]. Interestingly, across the reviews dedicated to acute effects on internal and external loads, the task constraints were always presented based on the format of play (smaller vs. larger formats and, sometimes, the influence of balanced vs. unbalanced formats) and pitch configuration (mainly the pitch size and the influence of different relative areas of play). Remaining constraints, such as the scoring method, action restrictions, or tactical/strategic mission, were also often reported, though with less emphasis. The training regimen was predominantly reported in the effects on the internal and external loads as well. A summary of evidence about the effects of these constraints on internal and external loads is provided in Table 3. Despite the relevance of task constraints in directly influencing internal and external load demands of SSGs, contextual factors, such as age group, expertise, competitive level, or maturation, seemed to also act as moderators [19, 28]. Surprisingly, fitness level was not tested as a possible covariable for physiological and physical responses in SSGs.

In the reviewed articles, it was observed that the main interactions tested between task constraints were between formats and relative area per player [26]. However, more studies should be conducted on the concurrent effects of different task constraints. Additionally, future systematic reviews should add sections discussing intra- and inter-individual variability during SSGs [33–35]. For that purpose, covariables must be added, and longer randomized studies must be promoted to describe how variable workload outcomes can be obtained during the same SSG. The influence of day of the week (and recovery status), readiness, and other important contextual factors should also be added for a better understanding of how and when to use specific SSGs in the weekly training schedule.

In summary (see Table 3), internal load tends to be greater when using smaller game formats (i.e., fewer players), larger pitches, limitations imposed on ball touches, special defensive demands or rules (e.g., man-to-man marking), and regular encouragement, while the inclusion of goalkeepers (soccer) usually reduces internal load. The effects of training regimen and contextual factors on the internal load produced by SSGs are either unknown or inconsistent. The results are similar for external load, with two relevant differences: (i) findings consistently suggest that intermittent training regimens produce greater loads than continuous regimens and (ii) greater competitive levels tend to increase external load during games.

## 4.2. Acute effects of SSGs: Technical/tactical dimensions

Among the included SR, four of them reported acute effects on both tactical and technical outcomes [10, 19, 21, 30], one exclusively focused on tactical behavior [22], and one examined only technical actions [36]. Regarding the technical actions, the great majority of the studies were conducted using an observational methodology (coding the actions). Meanwhile, for tactical behavior, two main approaches were performed (observational analysis–coding behaviors; position-based data analysis–using metrics).

Task constraints predominantly focused on format and pitch configuration. Other constraints, such as scoring method, action restrictions, or tactical instructions, were also considered, but to a lesser extent. Interestingly, tactical instructions were more widely reported in this section, possibly due to the more pronounced effects on behaviors and the collective organization of players. A summary of the main findings and their consistency is reported in

Table 3. In the particular case of technical and tactical actions, contextual factors, such as mental fatigue, expertise, or skill level, and age group, were explored and were consistently reported as determinants to justify variations in tactical behaviors and technical actions. Curiously, fitness status had no significant impact on technical actions and tactical behaviors [22, 36].

Due to the intimate relationship between behavioral and physical demands in the game, a greater exploration of the topic in the SR would be expected. However, it seems that such an approach has not been regularly conducted. More original works, such as ones combining analyses of tactical behavior, collective organization, and physical demands [11, 37], should be carried out. This would be helpful because a more robust overview of the effects of task constraints on all performance dimensions (e.g., tactical/technical, physical/physiological, psychological) would help to better identify the most adequate for specific situations.

In summary (see Table 3), it has been found that game format, pitch configuration, assigned tactical/strategic missions, and contextual factors (e.g., match status, expertise level) consistently affect acute assessments. Meanwhile, scoring method provides inconsistent results, and there is no evidence found regarding action restriction. As for the assessment of technical actions or skill, game format, pitch configuration, scoring method, action restriction, tactical/strategic mission, and contextual factors all affect acute assessments, especially concerning the frequency and success of the actions. Indeed, only one of the constraint categories (training regimen) failed to provide consistent effects.

## 4.3. Adaptations to SSG-based programs

The use of SSG-based programs to promote changes in physical performance, technical/tactical skills, and the psychological dimension was summarized in six articles. Two SRMA [23, 27] were specifically dedicated to identifying the influence in different performance measures and technical skills, while one SRMA [6] exclusively focused on the effects on aerobic performance compared to other training methods (conventional endurance), and another compared the effects of SSGs with other training methods on the change-of-direction ability [20].

Across the SRMA, the positive effects of SSG-based programs on aerobic performance were consistent, mainly emphasizing that there were no significant differences from other running-based methods [6, 23, 27]. Going into more detail, the majority of intervention studies using SSGs lasted six to eight weeks, with two to three sessions/week, using 2 vs. 2 to 4 vs. 4 formats performed four to seven times per session (repetitions), 2–4 minutes each repetition and with a work-to-rest ratio between 1:0.5 and 1:1 [6, 23, 27]. However, the evidence for positive effects on repeated sprint ability, jumping performance, linear sprinting, or change-of-direction ability was not so consistent and unequivocal as in the case of aerobic performance [23, 27]. This might create an opportunity for more research, mainly that combining SSG-based programs with other training methods. Some original research has tested the combination of SSGs and running-based, high-intensity interval training [38, 39] or even SSGs combined with strength training in the weight room [40]. However, studies combining SSGs and other training methods are recent and few, and so more consistent research should be performed.

Additionally, the use of an SSG-based training approach across the week needs to also be researched. In the reviews analyzed, this topic was not addressed. However, it seems important to analyze how the distribution of different SSGs across the week (not merely as training intervention) can be organized and implemented. The original cohort study conducted recently tried to demonstrate the effects of a weekly schedule based on SSGs [40]. However, more robust study designs should be developed to make comparisons with other interventions. Additionally, there is a lack of discussion and research in original works about the individual response profile to SSG-based programs [41]. The interaction of response to fitness and

tactical/technical baseline levels, competitive level, and remaining workload made across the week should be analyzed to gain a more specific idea about the impact of SSGs in players.

In summary (see Table 4), SSGs consistently and homogeneously produce improvements in aerobic capacity that are similar to those provided by more traditional aerobic training methods. However, neuromuscular adaptations, as measured by strength, speed, and agility tests, among others, are very heterogeneous, and the evidence currently suggests that a combination of SSGs with other, complementary training methods is warranted. Tactical and technical improvements seem to occur, but research on the topic is still scarce, especially given that two traditional arguments for the implementation of SSGs are a better interpretation of the game (with consequently better tactical performance) and promote execution opportunities that allow players to contextually improve their skills [42]. Finally, analyses of psychological constructs in the context of SSGs have been limited to enjoyment, which is highly restrictive and provides a unidimensional look into a multifactorial reality. Furthermore, associations or correlations between enjoyment and other psychological constructs are not always linear [43, 44], and complex relationships may have to be addressed, as they can mediate the experience.

## 4.4. Limitations of the current research on SSGs

The existing systematic reviews of research on SSGs have revealed some shortcomings (i.e., limitations that reduce the scope and trustworthiness of the existing body of knowledge). Noteworthy limitations include (i) the limited application of randomized experimental designs and over-reliance on observational studies; (ii) within experimental studies, an over-reliance on single group interventions and with few sessions, lacking comparative power; (iii) a lack of pre-registration of the review protocols, meaning that initial protocols may easily be changed *a posteriori* to better fit the emerging data; (iv) the overall poor methodological quality of reviews and meta-analyses as assessed through AMSTAR-2; (v) a focus on soccer by most of the research, with very little systematized knowledge being derived about the impact of SSGs in other sports; (vi) the fact that the known effects of manipulating task constraints are largely related to soccer and perhaps not generalizable to other sports; (vii) the almost complete neglect of psychological variables and their interaction with other variables; and (viii) the inter- and intra-individual variation in response to SSGs remaining unexplored.

## 4.5. Directions for future SSG research

In light of the limitations that were identified, research on SSGs could benefit from the following: (i) design experimental studies lasting 12 or more weeks with parallel non-randomized (e.g., teams from the same age group but from different clubs) groups (or, preferably, randomized groups); (ii) pre-register the full protocol, either for a trial or for a systematic review; (iii) expand research to different team sports and explore to what extent the same types of SSGs and manipulation of task constraints induce similar or distinct effects; (iv) focus more intensely on long-term effects, thereby providing more reliable indicators of learning; (v) explore how variables of different natures (e.g., physical vs. technical) interact in such experiments; (vi) incorporate several complementary psychological variables, both as primary outcomes (to assess the impact of SSGs on these variables) and mediators of other variables (e.g., physiological measures, changes in tactical behavior); and (vii) explore the role of inter- and intra-individual variability in response to similar SSGs.

Admittedly, a single study will not be able to address all of these aspects, and so compromises will have to be made. Furthermore, since age, expertise level, and other covariables will likely have a profound impact on the results, any generalizations of data from very specific samples should be made with great caution.

## 5. Conclusions

Small-sided games are widely considered a powerful pedagogical tool, and in the last decade, their role in improving physical and physiological parameters has also gained increased recognition. As of late, SR and SRMA on the subject have attempted to organize the existing knowledge and provide a state-of-the-art account of it. In turn, our umbrella review aimed to organize the sum knowledge of existing SR and SRMA. Methodologically, current SR and SRMA all present low or critically low quality as assessed through AMSTAR-2, meaning that new SR and SRMA should be conducted, but while adopting more rigorous methodological criteria. Thematically, most of what we now know is derived from soccer applications. Therefore, it is not clear if similar SSGs and associated manipulation of task constraints produce similar adaptations across other sports.

Notwithstanding, SSGs appear to present acute and chronic effects in tactical, technical, and physical dimensions and, therefore, constitute a powerful tool for learning and improving physical fitness. Differently structured SSGs (e.g., in terms of format, pitch configuration, training regimen, among other factors) produce distinct effects, but it remains to be analyzed how they interact with psychological factors and what roles inter- and intra-individual variability play. In conclusion, despite the promises associated with SSGs, better quality research with a broader scope needs to be conducted.

## Supporting information

**S1 Checklist.**
(DOC)

## Author Contributions

**Conceptualization:** Filipe Manuel Clemente.

**Formal analysis:** Filipe Manuel Clemente.

**Methodology:** Filipe Manuel Clemente, Hugo Sarmento.

**Validation:** Filipe Manuel Clemente.

**Writing – original draft:** Filipe Manuel Clemente, José Afonso, Hugo Sarmento.

**Writing – review & editing:** Filipe Manuel Clemente, José Afonso, Hugo Sarmento.

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
