## [Decision Letter · Decision Letter 0]

9 Dec 2020

PONE-D-20-29848

Small-sided games: An umbrella review of systematic reviews and meta-analyses

PLOS ONE

Dear Dr. Clemente,

Thank you for submitting your manuscript to PLOS ONE. After careful consideration, we feel that it has merit but does not fully meet PLOS ONE’s publication criteria as it currently stands. Therefore, we invite you to submit a revised version of the manuscript that addresses the points raised during the review process.

The referees did not demonstrate too much enthusiasm with this work but they found some merit. Therefore, I would like to give you the chance to perform substantail revisions before further consideration. However, please note that a second round does not mean that the manuscript would be finally considered for publication.

We look forward to receiving your revised manuscript.

Kind regards,

Daniel Boullosa

Academic Editor

PLOS ONE

Journal Requirements:

2.In your Data Availability statement, you have not specified where the minimal data set underlying the results described in your manuscript can be found. PLOS defines a study's minimal data set as the underlying data used to reach the conclusions drawn in the manuscript and any additional data required to replicate the reported study findings in their entirety. All PLOS journals require that the minimal data set be made fully available. For more information about our data policy, please see http://journals.plos.org/plosone/s/data-availability.

3.Thank you for stating the following in the Acknowledgments Section of your manuscript:

[Filipe Manuel Clemente: This work is funded by Fundação para a Ciência e

Tecnologia/ Ministério da Ciência, Tecnologia e Ensino Superior through national funds

and when applicable co-funded EU funds under the project UIDB/EEA/50008/2020.

Hugo Sarmento gratefully acknowledge the support of a Spanish government subproject

Integration ways between qualitative and quantitative data, multiple case development,

and synthesis review as main axis for an innovative future in physical activity and

sports research [PGC2018-098742-B-C31] (Ministerio de Economía y Competitividad,

Programa Estatal de Generación de Conocimiento y Fortalecimiento Científico y

Tecnológico del Sistema I+D+i), that is part of the coordinated project ‘New approach

of research in physical activity 28 and sport from mixed methods perspective

(NARPAS_MM) [SPGC201800X098742CV0]’. No other specific sources of funding

were used to assist in the preparation of this article.]

 [The funders had no role in study design, data collection and analysis, decision to publish, or preparation of the manuscript.]

Reviewers' comments:

Reviewer's Responses to Questions

**Comments to the Author**

1. Is the manuscript technically sound, and do the data support the conclusions?

Reviewer #1: Partly

Reviewer #2: Partly

2. Has the statistical analysis been performed appropriately and rigorously? 

Reviewer #1: N/A

Reviewer #2: N/A

3. Have the authors made all data underlying the findings in their manuscript fully available?

Reviewer #1: No

Reviewer #2: No

4. Is the manuscript presented in an intelligible fashion and written in standard English?

Reviewer #1: Yes

Reviewer #2: Yes

5. Review Comments to the Author

Reviewer #1: Journal: Plos One

Manuscript ID: PONE-D-20-29848

Title: Small-sided games: An umbrella review of systematic reviews and meta-analyses

Reviewer' Comments:

In this manuscript, the authors conducted an umbrella review to summarize the evidence and qualify the methodological quality of SR and SRMA published on small-sided games in team ball sports.

The manuscript is well written, easy to read but, in our opinion, provides only very few new findings on this area.

Main concern: In the current study the authors need to explain clearly why there is a need to review SR and SRMA as these kinds of studies are already conducted based on relevant scientific questions... Clearly, the introduction must demonstrate what the current study may added to our knowledge in this area? The authors stated that it is still not clear how the applications of SSGs should be framed into structured, longer-term periods or what their long-term adaptations may be... In our opinion, to respond to these questions there is a need of original studies and not review of SR and SRMA...

In the abstract, please rewrite the conclusion. Be more specific, what it can be concluded based on the results...

Please provide the hypothesis at the end of the introduction.

Check the reference 9 and 25, same authors...

Reviewer #2: The overall idea of this umbrella review is interesting, and the topic is relevant. The methods seem appropriate, yet the inclusion criteria should be more clear and transparent to increase reproducibility. However, the results are poorly summarized and discussed. In the results, the authors are mostly counting up how many studies evaluated each one of the outcomes, but they do not (or very briefly) provide any information on what the studies found. The discussion repeats the same problem. A more in-depth discussion of the main results should be given.

I have the feeling that there is a sentence missing between lines 90 and 91 explaining how/why is expected that the variations proposed by SSgs would elicit long-term "adaptations" and "enhance deep learning". Why, on a neurobiological basis, would they be different?

"Finally, an external expert has been contacted in order to verify the final list of references included in this umbrella review in order to understand if there was any study that was not detected through our research." Who was this external expert? This action, for sure, will do no harm to the study. Still, there is also little addiction from having an "external expert" reviewing the included studies. There is no guarantee that the external expert also hasn't a limited/biased view of the field. This feeling is even stronger when we have no idea of who this expert is. Anyway, as it seems, the expert review has not resulted in the inclusion of any additional articles.

The inclusion criteria should be more transparent and clarified. "did not include relevant data in SSGs and/or team ball sports;" What do you mean by "relevant data"? What were the included outcomes? This criteria is far away to be clear... educational? developmental? The authors should provide, explicitly, the outcomes considered for inclusion.

Reviews including all study designs were potentially eligible?

"The AMSTAR-2 it is a rating system that classifies all reviews' quality level into critically low, low, moderate and high. The system classifies 16 items. The quality of each eligible SR and SRMA was analyzed by two researchers (FMC and HS)

independently."

Not all readers are familiarized with the AMSTAR-2 tool. More details of the items should be given.

The results are, in my view, not giving any meaningful information. Just saying how many studies evaluated each topic does not provide a concise, informative, or comprehensive notion of what we know in the topic. The authors should provide a straightforward interpretation of the findings, giving effect sizes (when presented by SR or SRMA) or synthesizing the main conclusions and giving directions of what was found when all the available literature is pooled together.

Discussion follows the same issue. When discussing their findings, the authors are once again just saying what was done in the literature, and little attention was given to what was found, the meaning and limitations of the findings.

The number of studies assessing each outcome

6. PLOS authors have the option to publish the peer review history of their article (what does this mean?). If published, this will include your full peer review and any attached files.

Reviewer #1: No

Reviewer #2: No

---

## [Author Response · Author response to Decision Letter 0]

21 Dec 2020

Reviewer #1: Journal: Plos One

Manuscript ID: PONE-D-20-29848

Title: Small-sided games: An umbrella review of systematic reviews and meta-analyses

Reviewer' Comments:

In this manuscript, the authors conducted an umbrella review to summarize the evidence and qualify the methodological quality of SR and SRMA published on small-sided games in team ball sports.

The manuscript is well written, easy to read but, in our opinion, provides only very few new findings on this area.

AUTHORS: DEAR REVIEWER, THANK YOU SO MUCH. WE HAVE MADE CHANGES IN THE UMBRELLA REVIEW FOLLOWING YOUR SUGGESTIONS. THE CHANGES RELATED TO YOUR COMMENTS WERE HIGHLIGHTED IN GREEN. MOREOVER, WE HAVE ADDED A PARAGRAPH TO EXPLAIN THE REASONS WHY WE ARE DEVELOPING AN UMBRELLA REVIEW (Last two paragraphs of introduction). ADDITIONALLY, WE DO THINK THAT THIS UMBRELLA REVIEW WILL HELP TO PROVIDE A BROADER PICTURE ABOUT THE CURRENT RESEARCH IN SGGS AS WELL AS PRESENT THE WAY HOW SSGS REVIEWS HAVE BEEN CONDUCTING (Tables 1-2). NEW RESEARCH LINES WILL BE PROPOSED FROM THE UMBRELLA REVIEW (section 4.4.). ADDITIONALLY, THE METHODOLOGICAL ASSESSMENT ALLOWS IDENTIFYING NEEDS THAT FUTURE RESEARCH PROTOCOLS SHOULD COVER (sections 2.5. and 3.3.). ALL OF THESE OBJECTIVES CANNOT BE ACHIEVED BY CONDUCTING REGULAR SYSTEMATIC REVIEWS OR ORIGINAL ARTICLES THAT CANNOT COVER A BROADER SPECTRUM OF RESEARCH IN A TOPIC. THIS IS THE NEW PARAGRAPHS: 

Main concern: In the current study the authors need to explain clearly why there is a need to review SR and SRMA as these kinds of studies are already conducted based on relevant scientific questions... Clearly, the introduction must demonstrate what the current study may added to our knowledge in this area? The authors stated that it is still not clear how the applications of SSGs should be framed into structured, longer-term periods or what their long-term adaptations may be... In our opinion, to respond to these questions there is a need of original studies and not review of SR and SRMA...

AUTHORS: DEAR REVIEWER, THANK YOU. WE HAVE ADDED A NEW PARAGRAPH ABOUT THE CONTRIBUTION OF AN UMBRELLA REVIEW FOR THIS TOPIC. THIS PARAGRAPH CAN BE FOUND IN THE INTRODUCTION (LAST PARAGRAPH). THESE ARE THE NEW PARAGRAPHS: “The last two decades have shown a proliferation of publications of empirical studies, and in the last decade, systematic reviews (SRs) of small-sided games have emerged. The progressive number of SRs about the topic provides an opportunity to synthesize and summarize the main evidence from multiple research syntheses. In fact, despite using the same topic (SSGs) different methodological approaches conducts different findings in SRs. For that reason, it is important to assess the methodological quality of the current SRs about SSGs, as well as, provide an opportunity to synthesize the main evidence and the lack of evidence that must provide new lines for future research. An umbrella review may offer a rapid review of the evidence and provide an overall examination of the body of information that is available for the topic of SSGs (14). In fact, an umbrella review will provide a clear opportunity to have a wide picture of the consistency or not of evidence around SSGs (14). Additionaly, it will be possible to have a broader picture about different approaches conducted in SSGs.

Based on these reasons, the purposes of this article is to employ an umbrella review in SSGs that allows: (i) to systematically review available SR and systematic reviews with meta-analysis (SRMA) about SSGs in team ball sports; (ii) to qualify the methodological quality of these SR and SRMA, as well as to identify their strengths and limitations; and (iii) to summarize the main evidence presented in SR and SRMA, identify possible gaps in the literature, and provide recommendations for future research on SSGs. These objectives will help to provide a wide picture of the research conducted in SSGs, as well as define new lines and opportunities for research. Will also help to provide relevant information about the consistency or not of specific evidence about SSGs. For that reason, it is expectable that this umbrella review reveals consistency or inconsistency of changing task conditions in specific internal and external load demands, as well as, in technical/tactical skills by summarizing the SRs. Additionally, it is expectable to provide information about consistency or not of SSGs for developing specific physical qualities or technical/tactical skills, by summarizing the evidence of meta-analysis about the topic.”

In the abstract, please rewrite the conclusion. Be more specific, what it can be concluded based on the results...

AUTHORS: DEAR REVIEWER, THANK YOU. WE HAVE CHANGED THE CONCLUSIONS BASED ON YOUR RECOMMENDATION. THIS IS THE NEW CONCLUSION: “The current umbrella review allowed to identify that most of the systematic review and meta-analysis conducted in SSGs presents low methodological quality considering the standards. Most of the systematic reviews included in this umbrella revealed that task constraints significantly change the acute responses in exercise, while SSGs are effective in improving aerobic capacity. Future original studies in this topic should improve the methodological quality and improve the experimental study designs for assessing changes in tactical/technical skills”.

Please provide the hypothesis at the end of the introduction.

AUTHORS: DEAR REVIEWER, THANK YOU. WE HAVE ADDED EXPECTATIONS BASED ON THE OBJECTIVES OF THE STUDY (IN THE LAST PARAGRAPH OF INTRODUCTION).

Check the reference 9 and 25, same authors...

AUTHORS: DEAR REVIEWER, THANK YOU. WE HAVE CORRECTED THE REFERENCE.

Reviewer #2: The overall idea of this umbrella review is interesting, and the topic is relevant. 

AUTHORS: DEAR REVIEWER, THANK YOU FOR YOUR COMMENTS AND FEEDBACKS. WE HAVE CHANGED THE MANUSCRIPT FOLLOWING YOUR COMMENTS. THE CHANGES RELATED TO YOUR SUGGESTIONS WERE HIGHLIGHTED IN YELLOW (IN-TEXT).

The methods seem appropriate, yet the inclusion criteria should be more clear and transparent to increase reproducibility. 

AUTHORS: DEAR REVIEWER, THANK YOU. WE HAVE FOLLOWED YOUR SUGGESTIONS FOR IMPROVING THE TRANSPARENCY OF THE METHODS. THIS IS THE NEW PARAGRAPH: “The inclusion criteria for this umbrella review were as follows: (i) only SR or SRMA in SSGs (not limited to the type of study designs included in the SR or SRMA) in team ball sports; (ii) any SR or SRMA in SSGs that included outcomes related to an internal and external load (acute responses to SSGs), fitness variables (adaptations after a certain period of SSGs-based intervention) technical/tactical measures (acute responses or adaptations), psychological or pedagogical dimensions (acute responses or adaptations); and (iii) peer-reviewed SR and SRMA written in English that provided full-text.”

However, the results are poorly summarized and discussed. In the results, the authors are mostly counting up how many studies evaluated each one of the outcomes, but they do not (or very briefly) provide any information on what the studies found. 

AUTHORS: DEAR REVIEWER, THANK YOU. HOWEVER, WE DO BELIEVE THAT THE TABLES 2, 3 AND 4 CORRESPONDS TO THOSE EXPECTATIONS. IN FACT, TABLE 2 SUMMARIZE AND DETAILS THE MAIN FINDINGS OF EACH REVIEW. TABLES 3 AND 4 PROVIDE INFORMATION ABOUT THE CONSISTENCY OF THE FINDINGS ACROSS THE INCLUDED REVIEWS AND META-ANALYSIS. TABLES PROVIDED SUCH INFORMATION, WHILE THE DESCRIPTION OF RESULTS PROVIDE DIFFERENT ANALYSIS TO NOT REPEAT INFORMATION. THIS WAS OUR RATIONALE. TABLE 2 PRESENTS A SUMMARY OF THE MAIN FINDINGS FOR EACH SPECIFIC TOPIC PRESENTED IN THE SYSTEMATIC REVIEWS. TABLES 3 AND 4 SUMMARIZE THE MAIN FINDINGS FOR EACH TOPIC CONSIDERING ALL THE AVAILABLE (AND INCLUDED) SYSTEMATIC REVIEWS AND META-ANALYSIS.

The discussion repeats the same problem. A more in-depth discussion of the main results should be given.

AUTHORS: DEAR REVIEWER, THANK YOU. HOWEVER, SECTIONS 4.1., 4.2., 4.3. AND 4.4. CLEARLY SYNTHESIZE THE FINDINGS, EVEN PRESENTING THE CONSISTENTENCY OR NOT OF THE FINDINGS. IN PARTICULAR, SECTION 4.4., SYNTHESIZE THE FINDINGS, ALSO ADDING A PRACTICAL IMPLICATION. FINALLY, coherently with the PURPOSE OF THE UMBRELLA REVIEW, a DISCUSSION ABOUT FUTURE RESEARCH DIRECTIONS was PROVIDED, AIMING TO USE THIS ARTICLE AS A NEW MARK FOR FUTURE ORIGINAL STUDIES.

I have the feeling that there is a sentence missing between lines 90 and 91 explaining how/why is expected that the variations proposed by SSgs would elicit long-term "adaptations" and "enhance deep learning". Why, on a neurobiological basis, would they be different?

AUTHORS: DEAR AUTHOR, THANK YOU. WE HAVE CHANGED THE SENTENCE.

"Finally, an external expert has been contacted in order to verify the final list of references included in this umbrella review in order to understand if there was any study that was not detected through our research." Who was this external expert? This action, for sure, will do no harm to the study. Still, there is also little addiction from having an "external expert" reviewing the included studies. There is no guarantee that the external expert also hasn't a limited/biased view of the field. This feeling is even stronger when we have no idea of who this expert is. Anyway, as it seems, the expert review has not resulted in the inclusion of any additional articles.

AUTHORS: DEAR AUTHOR, THANK YOU. HOWEVER, THIS PROCESS (DOUBLE-CHECK OF AN EXTERNAL EXPERT) IS RECOMMENDED BY PRISMA GUIDELINES AND IS A REQUIREMENT OF THE ASSESSMENT ITEMS IN AMSTAR-2 (16). THE EXTERNAL EXPERT IS A UNIVERSITY PROFESSOR (PH.D., ASSISTANT PROFESSOR AND TEN YEARS OF PUBLICATIONS IN SSGS, WITH INTERNATIONAL PUBLICATIONS IN THE TOPIC) . WE HAVE NOT ATTEMPTED TO BIAS THE EXPERT ANALYSIS IN ANY WAY. INDEED, WE ONLY PROVIDED THE LIST OF ARTICLES WE HAD INCLUDED AND THE INCLUSION AND EXCLUSION CRITERIA (AS SUGGESTED ALSO IN COCHRANE’S GUIDELINES FROM 2019). HOWEVER, WE DID NOT INFORM THE EXPERT ON WHICH DATABASES WE CONSULTED OR ON THE SEARCH STRATEGIES APPLIED, SO THAT THE EXPERT COULD CONDUCT UNBIASED SEARCHES. SUCH INFORMATION WAS ADDED IN THE SECTION. 

The inclusion criteria should be more transparent and clarified. "did not include relevant data in SSGs and/or team ball sports;" What do you mean by "relevant data"? What were the included outcomes? This criteria is far away to be clear... educational? developmental? The authors should provide, explicitly, the outcomes considered for inclusion.

AUTHORS: DEAR REVIEWER, THANK YOU. WE HAVE DETAILED THE CRITERIA FOR THE OUTCOMES: ““The inclusion criteria for this umbrella review were as follows: (i) only SR or SRMA in SSGs (not limited to the type of study designs included in the SR or SRMA) in team ball sports; (ii) any SR or SRMA in SSGs that included outcomes related to an internal and external load (acute responses to SSGs), fitness variables (adaptations after a certain period of SSGs-based intervention) technical/tactical measures (acute responses or adaptations), psychological or pedagogical dimensions (acute responses or adaptations); and (iii) peer-reviewed SR and SRMA written in English that provided full-text.”

Reviews including all study designs were potentially eligible?

AUTHORS: DEAR AUTHOR, THANK YOU. WE HAVE ADDED THIS INFORMATION IN THE ELIGIBILITY CRITERIA: “(i) only SR or SRMA in SSGs (not limited to the type of study designs included in the SR or SRMA) in team ball sports”

"The AMSTAR-2 it is a rating system that classifies all reviews' quality level into critically low, low, moderate and high. The system classifies 16 items. The quality of each eligible SR and SRMA was analyzed by two researchers (FMC and HS)

independently."

Not all readers are familiarized with the AMSTAR-2 tool. More details of the items should be given.

AUTHORS: DEAR REVIEWER, THANK YOU. WE HAVE DETAILED THE 16 ITEMS. THE NEW PARAGRAPH IS “The system classifies 16 items, namely (17): (i) information about the use of PICO; (ii) statement about the methods made before conducting research; (iii) explanation for inclusion of study designs; (iv) use of comprehensive literature search strategy; (v) study selection made in duplicate; (vi) data extraction in duplicate; (vii) list of excluded studies and reasons; (viii) describe included studies in detail; (ix) assessing the risk of bias; (x) report sources of funding for the included studies; (xi) appropriate statistical methods used in the meta-analysis; (xii) assess the potential impact of risk of bias on the results; (xiii) consider the risk of bias in primary outcomes when interpreting/discussing the results; (xiv) appropriate explanation about heterogeneity observed in the results; (xv) conduct an adequate investigation of publication bias and discuss its likely impact on the results, and (xvi) report potential sources of conflict of interest.”

The results are, in my view, not giving any meaningful information. Just saying how many studies evaluated each topic does not provide a concise, informative, or comprehensive notion of what we know in the topic. The authors should provide a straightforward interpretation of the findings, giving effect sizes (when presented by SR or SRMA) or synthesizing the main conclusions and giving directions of what was found when all the available literature is pooled together.

AUTHORS: DEAR REVIEWER, THANK YOU. HOWEVER, WE DO BELIEVE THAT THE TABLES 2, 3 AND 4 CORRESPONDS TO THOSE EXPECTATIONS. IN FACT, TABLE 2 SUMMARIZE THE MAIN FINDINGS OF EACH REVIEW (EFFECT SIZE WERE INCLUDED IN THE NEW VERSION). TABLES 3 AND 4 PROVIDE INFORMATION ABOUT THE CONSISTENCY OF THE FINDINGS ACROSS THE INCLUDED REVIEWS AND META-ANALYSIS. TABLES PROVIDED SUCH INFORMATION, WHILE THE DESCRIPTION OF RESULTS PROVIDE DIFFERENT INFORMATION TO NOT REPEAT INFORMATION. THIS WAS OUR RATIONALE. 

Discussion follows the same issue. When discussing their findings, the authors are once again just saying what was done in the literature, and little attention was given to what was found, the meaning and limitations of the findings.

AUTHORS: DEAR REVIEWER, THANK YOU. WE HAVE ADDED A SUMMARY OF EVIDENCE IN THE END OF EACH SECTION OF DISCUSSION. PLEASE CONSIDER THE HIGHLIGHTED PARAGRAPHS IN SECTIONS 4.1, 4.2 AND 4.3. FOR SECTION 4.1.: “In summary (see Table 3), internal load tends to be greater when using smaller game formats (i.e., fewer players), larger pitches, limitations imposed on ball touches, special defensive demands or rules (e.g., man-to-man marking), and regular encouragement, while the inclusion of goalkeepers (soccer) usually reduces internal load. The effects of training regimen and contextual factors on the internal load produced by SSGs are either unknown or inconsistent. The results are similar for external load, with two relevant differences: (i) findings consistently suggest that intermittent training regimens produce greater loads than continuous regimens and (ii) greater competitive levels tend to increase external load during games.

FOR SECTION 4.2 “In summary (see Table 3), it has been found that game format, pitch configuration, assigned tactical/strategic missions, and contextual factors (e.g., match status, expertise level) consistently affect acute assessments. Meanwhile, scoring method provides inconsistent results, and there is no evidence found regarding action restriction. As for the assessment of technical actions or skill, game format, pitch configuration, scoring method, action restriction, tactical/strategic mission, and contextual factors all affect acute assessments, especially concerning the frequency and success of the actions. Indeed, only one of the constraint categories (training regimen) failed to provide consistent effects.

FOR SECTTION 4.3 “In summary (see Table 4), SSGs consistently and homogeneously produce improvements in aerobic capacity that are similar to those provided by more traditional aerobic training methods. However, neuromuscular adaptations, as measured by strength, speed, and agility tests, among others, are very heterogeneous, and the evidence currently suggests that a combination of SSGs with other, complementary training methods is warranted. Tactical and technical improvements seem to occur, but research on the topic is still scarce, especially given that two traditional arguments for the implementation of SSGs are a better interpretation of the game (with consequently better tactical performance) and promote execution opportunities that allow players to contextually improve their skills (42). Finally, analyses of psychological constructs in the context of SSGs have been limited to enjoyment, which is highly restrictive and provides a unidimensional look into a multifactorial reality. Furthermore, associations or correlations between enjoyment and other psychological constructs are not always linear (43,44), and complex relationships may have to be addressed, as they can mediate the experience.”

The number of studies assessing each outcome

AUTHORS: DEAR REVIEWER, THANK YOU. THESE NUMBERS CAN BE OBSERVED IN TABLE 1 AND ALSO IN SECTION 3.2.

---

## [Decision Letter · Decision Letter 1]

1 Feb 2021

Small-sided games: An umbrella review of systematic reviews and meta-analyses

PONE-D-20-29848R1

Dear Dr. Clemente,

We’re pleased to inform you that your manuscript has been judged scientifically suitable for publication and will be formally accepted for publication once it meets all outstanding technical requirements.

Kind regards,

Daniel Boullosa

Academic Editor

PLOS ONE

Additional Editor Comments (optional):

Reviewers' comments:

Reviewer's Responses to Questions

**Comments to the Author**

1. If the authors have adequately addressed your comments raised in a previous round of review and you feel that this manuscript is now acceptable for publication, you may indicate that here to bypass the “Comments to the Author” section, enter your conflict of interest statement in the “Confidential to Editor” section, and submit your "Accept" recommendation.

Reviewer #1: All comments have been addressed

Reviewer #2: All comments have been addressed

2. Is the manuscript technically sound, and do the data support the conclusions?

Reviewer #1: Yes

Reviewer #2: Partly

3. Has the statistical analysis been performed appropriately and rigorously? 

Reviewer #1: Yes

Reviewer #2: N/A

4. Have the authors made all data underlying the findings in their manuscript fully available?

Reviewer #1: (No Response)

Reviewer #2: Yes

5. Is the manuscript presented in an intelligible fashion and written in standard English?

Reviewer #1: Yes

Reviewer #2: Yes

6. Review Comments to the Author

Reviewer #1: The authors respond adequately to all our comments. Thus, I endorse the manuscript for publication in PlosOne.

Thank you and congratulations.

Reviewer #2: The authors are now presenting an improved version of the manuscript.

I have no further comments.

7. PLOS authors have the option to publish the peer review history of their article (what does this mean?). If published, this will include your full peer review and any attached files.

Reviewer #1: No

Reviewer #2: No

---

## [Editor Report · Acceptance letter]

3 Feb 2021

PONE-D-20-29848R1 

Small-sided games: An umbrella review of systematic reviews and meta-analyses 

Dear Dr. Clemente:

I'm pleased to inform you that your manuscript has been deemed suitable for publication in PLOS ONE. Congratulations! Your manuscript is now with our production department. 

Kind regards, 

on behalf of

Dr. Daniel Boullosa 

Academic Editor

PLOS ONE